

# Trait correlates of distribution trends in the Odonata of Britain and Ireland

Gary D. Powney[1], Steve S.A. Cham[2], Dave Smallshire[3] and Nick J.B. Isaac[1]

[1] Biological Records Centre, NERC Centre for Ecology & Hydrology, Wallingford, UK
[2] British Dragonfly Society, (BDS), Silsoe, UK
[3] Dragonfly Conservation Group, British Dragonfly Society, Chudleigh, UK

## ABSTRACT

A major challenge in ecology is understanding why certain species persist, while others decline, in response to environmental change. Trait-based comparative analyses are useful in this regard as they can help identify the key drivers of decline, and highlight traits that promote resistance to change. Despite their popularity trait-based comparative analyses tend to focus on explaining variation in range shift and extinction risk, seldom being applied to actual measures of species decline. Furthermore they have tended to be taxonomically restricted to birds, mammals, plants and butterflies. Here we utilise a novel approach to estimate occurrence trends for the Odonata in Britain and Ireland, and examine trait correlates of these trends using a recently available trait dataset. We found the dragonfly fauna in Britain and Ireland has undergone considerable change between 1980 and 2012, with 22 and 53% of species declining and increasing, respectively. Distribution region, habitat specialism and range size were the key traits associated with these trends, where habitat generalists that occupy southern Britain tend to have increased in comparison to the declining narrow-ranged specialist species. In combination with previous evidence, we conclude that the lower trend estimates for the narrow-ranged specialists could be a sign of biotic homogenization with ecological specialists being replaced by warm-adapted generalists.

# INTRODUCTION

Defaunation, the loss of species and populations (*Dirzo et al., 2014*), is occurring at an alarming rate with recent estimates suggesting that the current extinction rate is 1,000 times that of the historical natural background rate (*De Vos et al., 2014*). These declines are driven by environmental change, particularly habitat loss and climate change, and can be measured in a number of ways, e.g., changes in distribution and abundance (*Thomas et al., 2004*; *Biesmeijer et al., 2006*; *Butchart et al., 2010*; *Chen et al., 2011*). Variation in species responses to environmental change has been found across broad taxonomic groups (*Hickling et al., 2006*; *Angert et al., 2011*) but also within taxonomic groups, i.e., between species within an order (*Hickling et al., 2005*). A major challenge in conservation ecology is

Corresponding author
Gary D. Powney,
gary.powney@ceh.ac.uk

to gain a better understanding of this interspecific variation in response to environmental change, i.e., what enables certain species to persist while others decline?

Species traits play an important role in determining species' ability to resist environmental change. For example, several studies have shown that ecological generalists out-perform specialists in times of environmental change (*Walker & Preston, 2006*; *Ozinga et al., 2012*; *Newbold et al., 2013*). Such comparative trait-based analyses are popular, as the models help to identify the main drivers of change and allow the prediction of future biodiversity changes based on environmental forecasts (*Fisher & Owens, 2004*; *Cardillo et al., 2006*). Previous comparative trait analyses have tended to focus on explaining variation in range shift (*Angert et al., 2011*; *Mattila et al., 2011*; *Grewe et al., 2012*) and extinction risk (*Purvis et al., 2000*; *Koh, Sodhi & Brook, 2004*; *Cardillo et al., 2008*; *Cooper et al., 2008*; *Fritz, Bininda-Emonds & Purvis, 2009*). Despite its popularity, the comparative trait-based approach has seldom been applied to direct measures of species' changing status (i.e., rates of decline or increase). Currently data on such measures of decline are rare, particularly at large (e.g., national) scales and across multiple species. With the increase in public participation in biological recording, the availability of large-scale distribution datasets has increased (*Silvertown, 2009*). Such data tend to be collected without systematic protocols and thus contain many forms of sampling bias and noise, making it hard to detect genuine signals of change (*Tingley & Beissinger, 2009*; *Hassall & Thompson, 2010*; *Isaac et al., 2014*). However, recent advances in analytical approaches have improved our ability to estimate reliable trends from these unstructured biological records (*Isaac et al., 2014*). In this study, we utilise these novel approaches to estimate trends in occurrence for the Odonata in Britain and Ireland, and use species traits to test hypotheses for the interspecific variation in trends.

We chose to examine Odonata for a number of reasons. Firstly, previous trait-based comparative analyses have tended to focus on birds, mammals, plants and butterflies. Despite being highly species rich and their crucial role across ecosystems, the non-butterfly invertebrate fauna are comparatively poorly studied (*ICUN, 2001*; *Dirzo et al., 2014*). Secondly, Odonata are thought to be excellent bioindicators as they are sensitive to degradation of water ecosystems (*Samways & Steytler, 1996*; *Sahlén & Ekestubbe, 2001*; *Lee Foote & Rice Hornung, 2005*). Thirdly, they provide a valuable ecosystem service as they feed on many insect pests (*Brooks & Lewington, 2007*). Finally, the publication of a new atlas (*Cham et al., 2014*) and trait datasets (*Powney et al., 2014*) for British Odonata together constitute some of the best quality data of any non-butterfly invertebrate group. Previous research based on Odonata occurrence data has focussed on the impact of climate change on phenology and distribution. For example, *Hassall et al. (2007)* discovered that emergence from overwintering had significantly advanced over the past 50 years, while *Hickling et al. (2005)* showed that the upper latitudinal margin shifted north between 1960 and 1995. Outside Britain, *Bush et al. (2014)* used species distribution models (SDMs) to predict which Australian odonates were under threat from climate change.

Several studies have utilised traits to explain variation in several aspects of Odonata ecology, but typically focus on explaining variation in species response to climate change.

In terms of phenological advancement, *Hassall et al. (2007)* noted that spring species and those without egg diapause exhibited increased phenological shifts. *Angert et al. (2011)* examined trait correlates of range shift across multiple taxonomic groups, finding that exophytic Odonata species in Britain shifted further north, on average, than endophytic species. These insights, combined with extensive knowledge about their natural history (*Brooks & Lewington, 2007*), form the basis of seven competing hypotheses (outlined below) that aim to explain the interspecific variation in the distribution trends among British Odonata.

All traits included in the analysis have been shown to affect species' ability to respond to environmental change. Habitat breadth is frequently related to species trends, where habitat generalists outperform specialists due to their greater ability to adapt to novel environmental conditions (*Fisher & Owens, 2004*; *Menéndez et al., 2006*; *Botts, Erasmus & Alexander, 2012*). *Ball-Damerow, M'Gonigle & Resh (2014)* found evidence of the widespread expansion of habitat generalists which has led to biotic homogenization in the dragonfly fauna of California and Nevada over the last century. We test the hypothesis that Odonata in Britain and Ireland follow the patterns outlined above, with generalists out-performing specialists. Dispersal ability affects species' ability to respond to environmental pressures, with higher dispersal ability linked to an enhanced ability to respond (*Thomas et al., 2001*; *Pöyry et al., 2011*; *Grewe et al., 2012*). Using SDMs, *Hof et al. (2012)* found lentic (i.e., pond and lake dwelling) species had a greater ability to track changes in their climatic niche. This was linked to greater dispersal ability, which is essential given the ephemeral nature of their breeding sites (*Hof, Brandle & Brandl, 2006*). We predict lentic species will have higher (more positive) trend estimates than lotic species as their increased dispersal ability enables them to persist during times of environmental change through the efficient relocation to newly suitable areas. Geographic range size and body size are both frequently used as surrogates for a whole host of traits associated with ecological specialism and competitive ability (*Gittleman, 1985*; *Gaston, 2003*; *Angert et al., 2011*). We predict that widespread species and the larger, therefore more competitive species, are likely to show positive trends. Climate warming has increased the suitability of the landscape to those species that were previously limited by their lower thermal tolerance threshold (*Devictor et al., 2008*; *Dingemanse & Kalkman, 2008*; *Bellard et al., 2012*), and evidence of the loss of northern species has been seen in a variety of taxonomic groups across a variety of geographic regions (*Hill et al., 2002*; *Devictor et al., 2008*; *Myers et al., 2009*; *Foufopoulos, Kilpatrick & Ives, 2011*). We therefore predict that southerly distributed species will show the most positive trend estimates. An additional aspect of climate change that has been linked with trends in Odonata is the increase in flood events in Britain. Species which overwinter as larvae are particularly vulnerable to flooding as they can be swept away from ideal habitat areas to unsuitable regions in which they cannot persist (*Cham et al., 2014*). As a result, we predict species that overwinter as larvae will have undergone the greatest declines. Finally we test the hypothesis that flight period will be positively related with species' trend. *Grewe et al. (2012)* argued that species with longer

flight periods have increased dispersal ability, and therefore have a greater capacity to adapt in response to environmental change.

## MATERIALS & METHODS

### Occurrence trends

Trends were estimated from Odonata distribution records in Britain and Ireland collected by the Dragonfly Recording Network and coordinated by the British Dragonfly Society. Our analyses are based on 588,480 records of 36 native species collected between 1980 and 2012 where the recording date is known and the location was recorded to 1 km$^2$ precision or better. As these occurrence records were collected without a specific sampling design they contain a variety of bias which inhibit their use in estimating reliable trends. For example, the number of records collected each year has increased dramatically over time (*Cham et al., 2014*), such that simply counting the number of occupied sites would produce biased trend estimates (*Prendergast et al., 1993*; *Isaac et al., 2014*). To account for these biases we estimated species trends using an approach based on Bayesian occupancy modelling (*Van Strien, Van Swaay & Termaat, 2013*; *Isaac et al., 2014*). We first arranged the records into 212,574 visits, which were defined as unique combinations of date and 1 km$^2$ grid cell (site). For each visit, each of the 36 species was coded as either recorded (1) or not-recorded (0). We then selected sites with at least three years of data, ensuring we retained only the well-sampled sites (Fig. 1). Our final dataset contains 467,899 records from 157,507 visits to 11,435 sites (64,005 site-year combinations). We ran occupancy models for each species based on the methodology of *Van Strien, Van Swaay & Termaat (2013)* and *Isaac et al. (2014)*. The approach uses two hierarchically coupled sub-models, one, the state model, governs the true presence/absence of a species at a site in a given year, the second, the observation model, governs the probably of detecting that species given its presence or absence, and is therefore conditional on the state model. The detection probability per visit is a function of the number of species recorded on that visit (the 'list length': see Appendix S1 for detailed model description). For each site-year combination the model estimates presence or absence for the species in question given variation in detection probability: from this the proportion of occupied sites ('occupancy') was estimated for each year. Finally, within the Bayesian framework, a linear trend was fitted to these annual proportions to identify a temporal trend in species occupancy. The slope of this regression of occupancy against year was used as the species-specific trend measure in the cross-species comparative analysis.

### Species' trait data

We included data on seven traits extracted from *Powney et al. (2014)* (Table 1). Two traits were based on characteristics of a species' distribution pattern, the first, species status, was measured as an ordinal variable based on distribution size, moving from very rare through to very widespread. Secondly, distribution region was a categorical variable that defined a species broad climatic restriction, with species classified into one of four levels, northern, southern (which included continental species from the original classification), oceanic

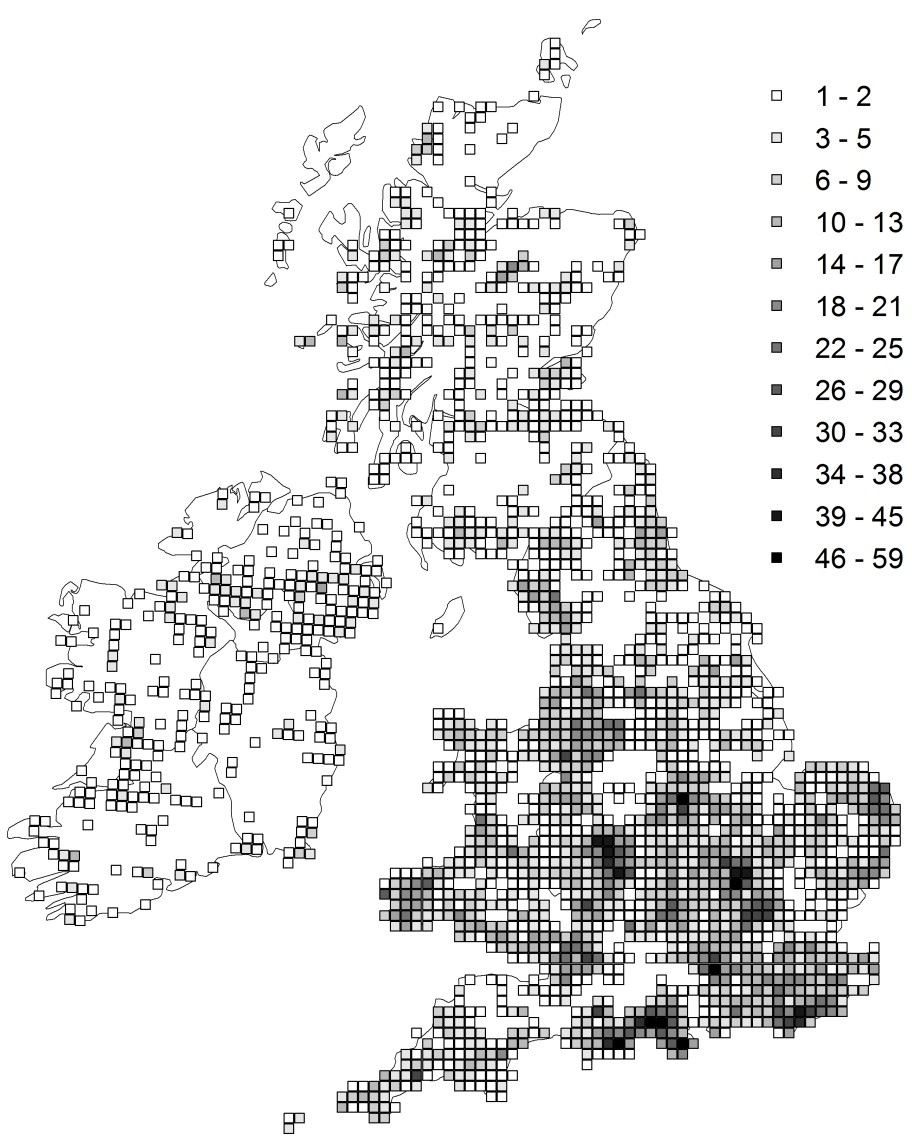

| | |
|---|---|
| ▫ | 1 - 2 |
| ▫ | 3 - 5 |
| ▫ | 6 - 9 |
| ◻ | 10 - 13 |
| ◻ | 14 - 17 |
| ◼ | 18 - 21 |
| ◼ | 22 - 25 |
| ◼ | 26 - 29 |
| ■ | 30 - 33 |
| ■ | 34 - 38 |
| ■ | 39 - 45 |
| ■ | 46 - 59 |

**Figure 1 The distribution and density of sites (monads) from which the trend estimates were derived.** The shading represents the number of unique sites within the hectad that were included in the analysis.

or widespread based on their distribution pattern. We included a single morphological trait, thorax length (mm), which was taken as the mean of multiple measurements from museum specimens. Flight period duration was measured as the number of months during which adults are typically recorded in flight. We included two habitat based traits, habitat breadth measured the number of broad habitats a species can utilise (maximum of 6), while breeding habitat classified species based on breeding habitat preference, lentic, lotic or both. Finally, we classified species based on their overwintering stage, either eggs, larvae or both. Distribution status was coded as an ordinal variable: very rare $= -1.5$, rare $= -1$, scarce $= -0.5$, local $= 0.5$, widespread $= 1$, very widespread $= 1.5$, and modelled as a

**Table 1** An overview of the Odonata traits included in the comparative analysis.

| Trait | Description | Class |
|---|---|---|
| Species status | Species categorised on distribution size: very widespread, widespread, local, scarce, rare, and very rare. | Ordinal |
| Distribution region | Broad climatic categorisation of species: widespread, southern, northern or oceanic. | Categorical |
| Thorax length | Mean thorax length based on 10 adult (5 male and 5 female) museum specimens (mm). | Continuous |
| Flight period | The duration of the flight period in months. | Continuous |
| Habitat breadth | A count of the number of habitat types utilised by the species. | Continuous |
| Breeding habitat | Species were classified on their preferred breeding habitat, either lentic, lotic or both. | Categorical |
| Overwint. stage | Species categorised as overwintering as larvae, eggs, or both. | Categorical |

continuous term in the analysis (as opposed to a factor). All continuous traits were centred on zero prior to the analysis.

## Comparative analysis

We used the *pgls* function from the R package *caper* (*Orme, 2012*) to run phylogenetically informed linear models to examine trait-trend relationships while accounting for phylogenetic non-independence (*Freckleton, Harvey & Pagel, 2002*). In all phylogenetically informed models, the level of phylogenetic correction (Pagel's λ) was estimated via maximum likelihood (*Pagel, 1999*; *Freckleton, Harvey & Pagel, 2002*). Due to data limitations, we used a phylogeny based on taxonomy for the analyses. The phylogeny was built using the *as.phylo* function from the R package *ape* (*Paradis, Claude & Strimmer, 2004*) with nodes based on Suborder, Family, Genus and Species, and all branch lengths were set to one.

We tested seven hypotheses about the drivers of species' trends whilst incorporating uncertainty in the trend estimates of each species. To do this, we fitted 10,000 trait-trend models: in each model we selected, at random, one value from the posterior distribution of trend estimates for each species. In all 10,000 models, we estimated the coefficients for each of the seven traits (described above) as fixed effects. From these models, we then calculated the mean and 95% confidence intervals for the trait-trend parameter estimates across all iterations.

## RESULTS

We found substantial variation in the trend estimates between species. Of 36 species included in the analysis, 8 had negative trends and 19 had positive trends where the 95 percentiles (2.5 and 97.5 percentiles) did not bridge zero (Appendix S2). Species that showed the greatest declines included: *Aeshna juncea* and *Sympetrum danae*, while *Anax imperator* and *Aeshna mixta* showed the greatest increases.

Key results from the comparative trait-analysis (Table 2 and Fig. 2) showed distribution status, habitat breadth and thorax length were positively associated with species trend, while species with longer flight periods tended to have lower trend estimates (i.e., they declined relative to species with short flight periods). Distribution region was an important predictor of species trend, where southern species increased relative to oceanic and widespread species. Notable exceptions to this trend include the declines in *Ischnura*

**Table 2 The mean and 95 percentiles of the trait coefficients estimated from 10,000 model iterations.** The coefficients for the categorical variables (overwintering stage, region and breeding habitat) are shown as contrasts to the reference category (eggs, southern and lentic, respectively). The mean level of phylogenetic signal (λ) across the 10,000 iterations is presented alongside its 95 percentiles.

| Parameter | Mean coef. | 95 percentile | |
| --- | --- | --- | --- |
| | | 0.025 | 0.975 |
| Thorax length | $4.87 \times 10^{-4}$ | $4.15 \times 10^{-4}$ | $5.65 \times 10^{-4}$ |
| Overwintering stage—both | $-0.005$ | $-0.006$ | $-0.004$ |
| Overwintering stage—larvae | $-2.82 \times 10^{-4}$ | $-9.58 \times 10^{-4}$ | $6.78 \times 10^{-4}$ |
| Flight period duration | $-0.004$ | $-0.005$ | $-0.002$ |
| Distribution status | $0.003$ | $0.002$ | $0.003$ |
| Region—northern | $0.003$ | $0.001$ | $0.004$ |
| Region—oceanic | $-0.007$ | $-0.007$ | $-0.006$ |
| Region—widespread | $-0.005$ | $-0.007$ | $-0.005$ |
| Habitat breadth | $0.001$ | $0.001$ | $0.001$ |
| Breeding habitat—both | $0.003$ | $0.002$ | $0.003$ |
| Breeding habitat—lotic | $0.004$ | $0.003$ | $0.004$ |
| λ (phylo. signal) | $0.035$ | $<0.001$ | $0.36$ |

*pumilio* and *Gomphus vulgatissimus* both of which were classified as southern species. Northern species showed the largest increases, although this result is highly uncertain (reflected in the wide 95% CI across the 10,000 iterations). Lentic species tended to have lower trend estimates than lotic species and those species that utilise both breeding habitat strategies. Finally, there appeared to be little evidence of an influence of overwintering stage on species trend. Trends tended to be similar between species that overwinter as eggs and those that overwinter as larvae. Species that can overwinter as both eggs and larvae had the lowest average trend estimates, however this category was comprised of four species only. In general, the phylogenetic signal across the model iterations was low, with a mean of 0.035 (0.0–0.36).

## DISCUSSION

We found that the dragonfly fauna in Britain and Ireland has undergone considerable change during recent decades, with high levels of interspecific variation in occurrence trends. We found 8 species (22%) had declined, whereas 19 species (53%) showed increasing trends. The large number of species with positive trends is likely to reflect the recovery of dragonfly populations in response to increased water quality in Britain since the mid-20th century. Although a greater number of species increasing than declining is good news for conservation, this could reflect biotic homogenization, i.e., the fauna becoming dominated by a certain group of species, leading to the erosion of local and regional difference between communities (*Keith et al., 2009*).

We found distribution region was a key correlate of Odonata occurrence trends, with southern species tending to have higher trend estimates than the oceanic and widespread species (Fig. 2). This result is in line with our hypothesis that increased temperatures has increased the climate suitability of Britain and Ireland for southerly distributed species.
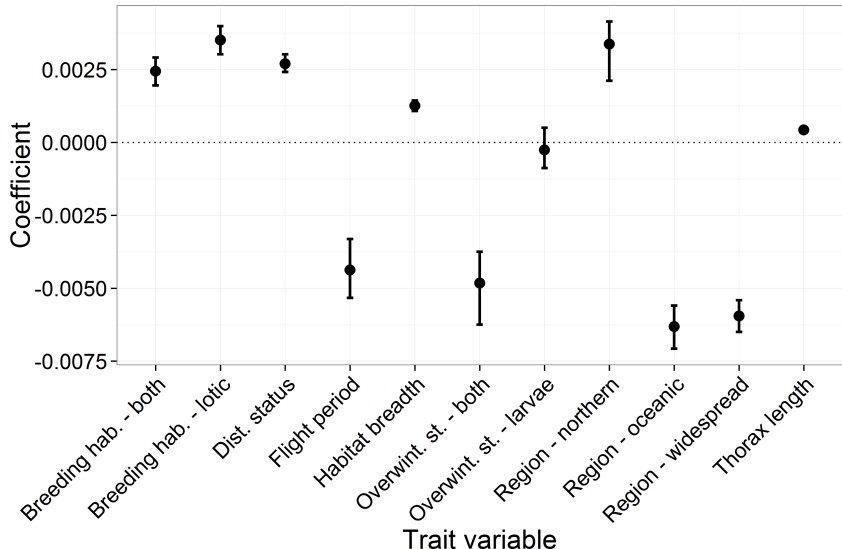

**Figure 2 The mean and 95 percentiles of the trait coefficients across 10,000 model iterations.** Each categorical variable had a reference category which had a parameter estimate set to 0. The reference categories were as follows: region, "southern"; breeding habitat, "lentic"; and for overwintering stage, "eggs."

A variety of studies have provided evidence of this relationship, i.e., *Devictor et al. (2008)* found bird communities in France between 1989 and 2006 were increasingly dominated by species that prefer warmer conditions, while *Lima et al. (2007)* found evidence of northward range expansions in warm-water adapted Portuguese algae. *Hickling et al. (2005)* used distribution region to explain variation in range shift and expansion in British Odonata, finding that southern species showed greater poleward shifts and expansions compared to northern species. We found little evidence of a difference between northern and southern species, however this is likely due the low number of northern species ($n = 4$) included in our dataset. Interestingly, *Angert et al. (2011)* found no correlation between range shift and position of the northern range limit (which is related to our measure of distribution region). Despite the wealth of evidence that points to climate change as the likely driver of increases in southern species, we cannot ignore the role of improved water quality and standing water availability in southern Britain (*Hickling et al., 2005*; *Vaughan & Ormerod, 2012*; *Cham et al., 2014*). *Durance & Ormerod (2009)* noted that improved water quality can confound attempts to detect the impact of climate change on freshwater macroinvertebrates. Southern species are likely to have benefitted from both the increased water quality in southern Britain and improved climate suitability, while the former was the main positive driver for widespread species, this could explain the greater increases in southern compared to widespread species. Not all southern species showed positive trends (notably *Ischnura pumilio* and *Gomphus vulgatissimus*); here the limited expansion is likely due to a lack of suitable habitat.

Numerous studies have related habitat breadth to species trends and tend to find that habitat generalists outperform specialists (*Fisher & Owens, 2004*; *Menéndez et al., 2006*;

*Botts, Erasmus & Alexander, 2012*). Much of the evidence of this relationship is based on studies of terrestrial organisms (*Biesmeijer et al., 2006*; *Ozinga et al., 2012*; *Newbold et al., 2013*), with a notable exception from *Ball-Damerow, M'Gonigle & Resh (2014)*. Here, we found that the relationship holds in the Odonata fauna of the UK as habitat breadth was positively correlated with occurrence trend. The likely cause of this relationship is that habitat generalists have a greater ability to adapt to novel environmental conditions, which is particularly important in our current climate of anthropogenically driven environmental change (*Travis, 2003*; *Newbold et al., 2013*). Thorax length and distribution status (used here as a measure of range size) were positively related to occurrence trends, i.e., narrow ranged, "rare," small sized species tended to have lower trend estimates than wider ranging, larger species. Geographic range size and thorax length are often used as surrogates for traits associated with ecological specialism and competitive ability (*Gittleman, 1985*; *Gaston, 2003*; *Angert et al., 2011*). Therefore, as with habitat specialism above, we believe this result is driven by the greater ability of competitive ecological generalists to adapt to environmental change than specialists.

*Hof et al. (2012)* found lentic (i.e., pond and lake dwelling) species had a greater ability to track changes in their climatic niche due to their greater dispersal ability, essential given the ephemeral nature of their breeding sites. We hypothesised that the greater dispersal ability of lentic species would promote their resilience to environmental change leading to a higher average trend estimate than lotic species. The results in this study do not support our hypothesis as lentic species tended to have lower trend estimates than lotic species. Differences in mean trend between lentic and lotic species are likely due to differences in the impact environmental stressors (e.g., climate change, eutrophication and other forms of habitat degradation), interactions between them and subsequent restoration between rivers and lakes (*Vaughan & Ormerod, 2012*). A study aimed at improving our understanding of the variation between lentic and lotic species is a prime candidate for future work.

Finally, we found that flight period was negatively related to occurrence trend, a result contrary to expectations. As with the lentic/lotic hypothesis, initially we suspected that species with greater dispersal ability would show higher trends as increased dispersal capacity increases the ability to mitigate the negative effects of environmental. We used flight period as a surrogate of dispersal ability on the premise that the longer the flight season the more time a species has to disperse (*Grewe et al., 2012*). It is plausible that the use of a more direct measure of dispersal ability would have produced a result that is consistent with the literature on dispersal ability and species trends. It is worth noting that the reliability of the trait-trend results depend on the accuracy of the underlying trait and trend data, and we note that within a given species, traits can vary spatially. One such plastic trait includes flight period that has been shown to vary with latitude (*Corbet, 2004*). Summarising this variation into a single value per trait per species is a common approach but can create noise in model results.

In conclusion, we found that a large number of dragonfly species have increased in the UK between 1980 and 2012, and is likely a response to increased water quality. We found

that habitat generalists that occupy southern Britain tend to have increased in comparison to the narrow-ranged specialist species of dragonfly. We believe this reflects the impact of environmental change, particularly climate change, as the increased ambient temperature in Britain and Ireland better suits species that are adapted to warmer conditions. The lower trend estimates for specialist species is a cause of conservation concern as this result combined with evidence in previous studies could be a sign of biotic homogenization with ecological specialists being replaced by warm-adapted generalists.

## ACKNOWLEDGEMENTS

We are indebted the British Dragonfly Society and its vast collection of volunteer recorders, without them this project would not be possible. We thank Oliver Pescott, Colin Harrower, Tom August and Louise Barwell for their advice on the data analysis. We thank Christopher Hassall and Lester Yuan for providing valuable feedback on an earlier draft of this study.

### Funding

This work was funded by the Natural Environment Research Council (NERC). The funders had no role in study design, data collection and analysis, decision to publish, or preparation of the manuscript.

### Grant Disclosures

The following grant information was disclosed by the authors:
Natural Environment Research Council (NERC).

### Competing Interests

Gary D. Powney and Nick J.B. Isaac are employees of NERC Centre for Ecology & Hydrology. Steve Cham and Dave Smallshire are members of the British Dragonfly Society.

### Author Contributions

- Gary D. Powney conceived and designed the experiments, performed the experiments, analyzed the data, contributed reagents/materials/analysis tools, wrote the paper, prepared figures and/or tables, reviewed drafts of the paper.
- Steve S.A. Cham and Dave Smallshire performed the experiments, contributed reagents/materials/analysis tools, reviewed drafts of the paper.
- Nick J.B. Isaac conceived and designed the experiments, performed the experiments, analyzed the data, contributed reagents/materials/analysis tools, wrote the paper, reviewed drafts of the paper.

### Data Availability

The research in this article did not generate any raw data.

## Supplemental Information

Supplemental information for this article can be found online at http://dx.doi.org/10.7717/peerj.1410#supplemental-information.

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

# PeerJ

**Paradis E, Claude J, Strimmer K. 2004.** APE: analyses of phylogenetics and evolution in R language. *Bioinformatics* **20**:289–290 DOI 10.1093/bioinformatics/btg412.

**Powney GD, Brooks SJ, Barwell LJ, Bowles P, Fitt RNL, Pavitt A, Spriggs R, Isaac NJB. 2014.** Morphological and geographical traits of the British Odonata. *Biodiversity Data Journal* **2**:e1041 DOI 10.3897/BDJ.2.e1041.

**Pöyry J, Leinonen R, Söderman G, Nieminen M, Heikkinen RK, Carter TR. 2011.** Climate-induced increase of moth multivoltinism in boreal regions. *Global Ecology and Biogeography* **20**:289–298 DOI 10.1111/j.1466-8238.2010.00597.x.

**Prendergast JR, Wood SN, Lawton JH, Eversham BC. 1993.** Correcting for variation in recording effort in analyses of diversity hotspots. *Biodiversity Letters* **1**:39–53 DOI 10.2307/2999649.

**Purvis A, Gittleman JL, Cowlishaw G, Mace GM. 2000.** Predicting extinction risk in declining species. *Proceedings of the Royal Society B—Biological Sciences* **267**:1947–1952 DOI 10.1098/rspb.2000.1234.

**Sahlén G, Ekestubbe K. 2001.** Identification of dragonflies (Odonata) as indicators of general species richness in boreal forest lakes. *Biodiversity and Conservation* **10**:673–690 DOI 10.1023/A:1016681524097.

**Samways MJ, Steytler NS. 1996.** Dragonfly (Odonata) distribution patterns in urban and forest landscapes, and recommendations for riparian management. *Biological Conservation* **78**:279–288 DOI 10.1016/S0006-3207(96)00032-8.

**Silvertown J. 2009.** A new dawn for citizen science. *Trends in Ecology & Evolution* **24**:467–471 DOI 10.1016/j.tree.2009.03.017.

**Thomas CD, Bodsworth EJ, Wilson RJ, Simmons AD, Davies ZG, Musche M, Conradt L. 2001.** Ecological and evolutionary processes at expanding range margins. *Nature* **411**:577–581 DOI 10.1038/35079066.

**Thomas JA, Telfer MG, Roy DB, Preston CD, Greenwood JJD, Asher J, Fox R, Clarke RT, Lawton JH. 2004.** Comparative losses of British butterflies, birds, and plants and the global extinction crisis. *Science* **303**:1879–1881 DOI 10.1126/science.1095046.

**Tingley MW, Beissinger SR. 2009.** Detecting range shifts from historical species occurrences: new perspectives on old data. *Trends in Ecology & Evolution* **24**:625–633 DOI 10.1016/j.tree.2009.05.009.

**Travis JMJ. 2003.** Climate change and habitat destruction: a deadly anthropogenic cocktail. *Proceedings of the Royal Society B—Biological Sciences* **270**:467–473 DOI 10.1098/rspb.2002.2246.

**Van Strien AJ, Van Swaay CAM, Termaat T. 2013.** Opportunistic citizen science data of animal species produce reliable estimates of distribution trends if analysed with occupancy models. *Journal of Applied Ecology* **50**:1450–1458 DOI 10.1111/1365-2664.12158.

**Vaughan IP, Ormerod SJ. 2012.** Large-scale, long-term trends in British river macroinvertebrates. *Global Change Biology* **18**:2184–2194 DOI 10.1111/j.1365-2486.2012.02662.x.

**Walker KJ, Preston CD. 2006.** Ecological predictors of extinction risk in the flora of lowland England, UK. *Biodiversity and Conservation* **15**:1913–1942 DOI 10.1007/s10531-005-4313-4.