# Peer review of "Trait correlates of distribution trends in the Odonata of Britain and Ireland"

_PeerJ, doi:10.7717/peerj.1410_

## Round 0.1 · original submission · Major Revisions

· Academic Editor

Major Revisions

The reviewers and I appreciate the work you have accomplished. However, Reviewer 2 has raised some major concerns surrounding the experimental design and the validity of findings. Reviewer 1 has raised some relatively minor issues. Please also note the annotated document, uploaded by Reviewer 1. I invite you to revise your paper in the light of all the comments made, and I emphasise that you will need to produce a convincing response to the concerns expressed by Reviewer 2.

·

Basic reporting

The basic reporting of the manuscript is fine. The text is written clearly (I have highlighted some minor typos in the attached annotated PDF) and the figures are generally well formatted (again, see a comment about Figure 2 in the attached PDF). No further comments.

Experimental design

The authors define a series of hypotheses concerning the traits that influence species trends in UK Odonata and test those hypotheses using novel trends databases that they authors themselves have compiled. The study is well-designed and carried-out. The methods are clear and reproducible with reference to work published elsewhere by the same authors.

Validity of the findings

Generally the interpretation of the results is sound. The main point of contention is the explanatory power of the trait models themselves and how to interpret the results in light of that low explanatory power. Clearly the null model falls within a delta-AIC of 2 of the top model, suggesting that there is little additional explanatory power by adding any traits to the models. A complementary issue is the nature of the "Distribution" trait, describing the general distribution type of the species. While the authors' own work (Powney et al., 2014) defines "Climatic restrictions" in terms of five categories, the current paper only uses four (omitting "Continental" as a classification). Since "Distribution" appears necessary for any model to outperform the null model, this is a little bit concerning. Either Powney et al. initially defined the trait incorrectly, or this analysis has applied the trait incorrectly, but either way there needs to be some clarification.

Powney G, Brooks S, Barwell L, Bowles P, Fitt R, Pavitt A, Spriggs R, Isaac N (2014) Morphological and Geographical Traits of the British Odonata. Biodiversity Data Journal 2: e1041. doi: 10.3897/BDJ.2.e1041

Comments for the author

This is a much-needed analysis, and is performed well by the authors. There is nothing wrong with the methods, as far as I can see. The only issue is in the interpretation of the results. I have added some additional comments to the attached PDF which suggest (i) additional figures, (ii) additional references, and (iii) some corrections to the writing.

·

Basic reporting

No comment

Experimental design

This manuscript describes an analysis of trends in occurrence of Odonates in publicly collected data. Analysis of this taxonomic group can potentially further document important changes in species occurrences from human activities, and as such, this work can provide a valuable contribution to the literature. I had several general comments regarding the statistical analysis, and a few additional specific comments that I may help improve the manuscript.

Data: The description of the data set is generally complete, but I wondered whether a subset of locations were sampled more than once per year? If so, how were data collected from these revisits summarized, and how were the biases associated with different numbers of samples per year addressed in the statistical analysis?
Since temperature and climate change are noted in the title and appear to be a main objective of this paper, it would be very useful to examine a trait that directly quantifies the temperature tolerance of different species. Is this type of data available for Odonates?

Statistical analyses: It seems that van Strien et al. (2010) defined “comprehensive species lists” as >3 species, whereas the current analysis defines it as greater than or equal to 3. Was there a reason for the different threshold in the number of observed species? The authors note that a different threshold of 2 species was also considered, and the results did not differ substantially. Was the same analysis run with > 3 species?
Other analyses of the publicly collected data have used analyses that distinguished between occurrence probability and detection probability, using repeated samples collected during the same year and at the same site to estimate detection probability. Was this type of analysis not possible with the present data set? What are the likely effects of differences in detection probability on the observed trends?
The formulation of the mixed model specifies year as a fixed effect and allows only a random intercept for site. So, the model forces the temporal trends at all sites to be the same, a restriction that eliminates the possibility of range shifts (e.g., a downward trend at one edge of the range, and an upward trend at the other edge). Allowing for a random contribution to the temporal trend term would seem to allow for this possibility. Was there a reason for excluding this possibility with the model formulation? Do other exploratory analyses seem to indicate that the effects of year are best modeled with the single fixed effect?

Validity of the findings

Results: The analyses of other responses (weighted, z-scores, etc.) is only mentioned in passing in the results section, and the manuscript would be strengthened by synthesizing the results over all of these responses. For example, the observation that northern Odonates decline relative to southern Odonates is not supported when one considers the results of all of the responses. Indeed, as far as I can tell, the only consistently significant response was the difference between widespread and southern species.
Since distribution is the primary predictor variable, it would be helpful to plot the values of temporal trends by distribution type. The exceptions to the overall trend noted in the text would then be easier for the reader to visualize.

Discussion: Given the different results that appear consistently across all responses (i.e., southern vs. widespread rather than southern vs. northern), the 2nd and 3rd paragraph of the discussion need revising. In particular, I was interested in how the authors interpret the apparent decreases in widespread species with regard to the warming climate. The 3rd paragraph focuses on northern species which do not appear to have consistent responses across the different response types, so some revision is necessary here as well.

Comments for the author

More specific comments by line:
Line 107: Define SDM before using the abbreviation.
Line 180-181: I think you mean that ordinal variables were modeled as categorical variables? If they were modeled as continuous variables, the numbers assigning different samples to categories would be interpreted as actual numbers, which is not what your results show.
Line 194: “year slope estimate”: many nouns strung together here. Edit to improve clarity.
Line 198: Would be good to explicitly state how z-score was defined.
Line 206-207: How was importance calculated?
Line 233: “Large proportion <of> declining species”
Line 290: Careful with the wording “in response to climate warming”. What started out as an observation regarding the consistency of observed trends with a hypothesis at the beginning of the paragraph turned into a much more strongly stated conclusion at the end of the paragraph.
Line 317: “don’t” should be “do not”.

---

## Round 0.2 · Minor Revisions

· Academic Editor

Minor Revisions

Thank you for addressing the specific comments made by the two reviewers. The manuscript has been considerably improved. However, in the second round of the reviewing process, some interesting points were raised by the reviewer. Please take these suggestions into consideration and supply a rebuttal.

I look forward to receiving a revised version which then might be acceptable for publication.

·

Basic reporting

No comments

Experimental design

No comments

Validity of the findings

No comments

Comments for the author

Review: Trait correlates of distribution trends in the Odonata of Britain and Ireland

I commend the authors for incorporating a model for detection probability in their revised analysis. I think this approach provides a more robust and accurate estimate of occupancy across the study area. However, I identified three major issues with the statistical analysis that may affect the validity of the results.

1. Categorical vs. continuous variables: The majority of the trait variables are categorical (with the exception of thorax length and perhaps habitat breadth) but the treatment of these variables in the analysis requires more explanation. A number of variables were described as being “coded as continuous variables” (Line 156), but this coding, combined with the linear model, suggests an initial assumption regarding the association between traits and trends. For example, overwintering stage is coded as egg = -1, both = 0, and larvae = 1. Fitting a linear relationship to this trait requires the mean trend for species that overwinter both as larvae and egg (coded as 0) to be between the trends observed for eggs and for larvae. Please clarify whether this is the hypothesized association. Conversely, leaving these traits coded as categorical values would require only one more degree of freedom to model, but allow a broader diversity of relationships. Also, what does the statement “ordinal variables were treated as continuous” (Line 159) mean?

2. Estimating the trend: The effect of year is taken into account in the Bayesian model as a random effect bt. If the aspect of interest in this analysis the yearly trend, why not model the year effect directly as a linear function (a + b*year) and add a random effect that accounts for residual variance about the annual trend. As currently formulated, I’m guessing that bt is drawn from a normal distribution, which doesn’t sound like an appropriate model for the year effect.

3. Trait-trend models:
a. I am guessing that separate models for each trait was fit to the estimated trends because the number of samples (N = 36 species) is far too small to fit a single model for all of the traits. If my guess is correct, please include this point in the methods.
b. Computing average p-values and average R2 values across each of the 10,000 models is not meaningful. To assess the significance of each coefficient, simply report statistics on the distribution of the 10,000 samples for each coefficient. If the bulk of the distribution is either greater than or less than zero, then we conclude that the coefficient is different from zero. The spreads of the error bars in Figure 2 seem too narrow to be representing the full distribution, so I suspect that a standard error on the mean (normalized by the square root of the number of samples) was computed, which would not be correct.
c. The best way to estimate the trait-trend model would be to incorporate it in the Bayesian model for trend, and directly compute a posterior distribution for each of the coefficients associated with the trait-trend relationship. This would of course increase the size of the model by a factor of 36, which may not be feasible for the computer resources you have available.

4. Other minor comments:
a. Line 147: “species broad climatic restriction” What does this mean?
b. Line 152-153: Please report how the range of values for habitat breadth. Is this truly continuous, or is this a very small number of possible values?

---

## Round 0.3 · accepted · Accept

· Academic Editor

Accept

I feel that you have successfully addressed the concerns of the reviewers and I am happy to accept this revised version now.
However, in proof-stage can you please change a very minor thing: Foufopoulos et al. published their paper in 2011, not in 2010 (as indicated in the reference list).